# Exploring the Effects of EEG-Based Alpha Neurofeedback on Working Memory Capacity in Healthy Participants

**DOI:** 10.3390/bioengineering10020200

**Published:** 2023-02-03

**Authors:** Rab Nawaz, Guilherme Wood, Humaira Nisar, Vooi Voon Yap

**Affiliations:** 1Department of Electronic Engineering, Faculty of Engineering and Green Technology, Universiti Tunku Abdul Rahman, Kampar 31900, Malaysia; 2Biomedical Engineering Research Division, University of Glasgow, Glasgow G12 8QQ, UK; 3Department of Psychology, University of Graz, Universitaetsplatz 2/III, 8010 Graz, Austria; 4BioTechMed-Graz, 8010 Graz, Austria; 5Centre for Healthcare Science and Technology, Universiti Tunku Abdul Rahman, Sungai Long 31900, Malaysia; 6Department of Computer Science, Aberystwyth University, Penglais SY23 3FL, UK

**Keywords:** EEG, neurofeedback training, working memory capacity, functional connectivity, N-back

## Abstract

Neurofeedback, an operant conditioning neuromodulation technique, uses information from brain activities in real-time via brain–computer interface (BCI) technology. This technique has been utilized to enhance the cognitive abilities, including working memory performance, of human beings. The aims of this study are to investigate how alpha neurofeedback can improve working memory performance in healthy participants and to explore the underlying neural mechanisms in a working memory task before and after neurofeedback. Thirty-six participants divided into the NFT group and the control group participated in this study. This study was not blinded, and both the participants and the researcher were aware of their group assignments. Increasing power in the alpha EEG band was used as a neurofeedback in the eyes-open condition only in the NFT group. The data were collected before and after neurofeedback while they were performing the N-back memory task (N = 1 and N = 2). Both groups showed improvement in their working memory performance. There was an enhancement in the power of their frontal alpha and beta activities with increased working memory load (i.e., 2-back). The experimental group showed improvements in their functional connections between different brain regions at the theta level. This effect was absent in the control group. Furthermore, brain hemispheric lateralization was found during the N-back task, and there were more intra-hemisphere connections than inter-hemisphere connections of the brain. These results suggest that healthy participants can benefit from neurofeedback and from having their brain networks changed after the training.

## 1. Introduction

Neurofeedback training (NFT) is an operant conditioning procedure that can modulate brain electrical activity such that one can learn to control one’s own activity. In late 1950s and early 1960s, Joe Kamiya was the first to demonstrate the ability of brainwaves through feedback and is, therefore, known as the father of neurofeedback. Barry Sterman discovered the clinical potential of NFT in the early 1970s and evaluated, for the first time, the association between brain activity and operant conditioning procedure by training cats to increase their sensorimotor rhythm (SMR) [1], which is usually considered a part of the alpha EEG rhythm [2]. During NFT, through a real-time interface with a computer, information about one’s brain activity is received as feedback and the desired modulation of brain activity leads to cognitive function, mood, motor function, and behavioural improvements [2,3].

Characterized by individual differences, an alpha band NFT design is one of the most effective protocols [4] and its functional role in neurofeedback (NF) protocols of cognitive performance has been increasingly discussed in the literature [3,5]. Over the past two decades, research has reported a relationship between alpha activity and memory functions [6,7] and has shown that alpha and theta activities are related to memory performance [8]. However, alpha reactivity and event-related changes in alpha power also show that, during actual task demands, the cognitive performance (in particular memory performance) increases with the suppression of alpha activity [8]. Memory processing is also associated with pre-stimulus alpha EEG activity [9]. Similarly, alpha NFT has been shown to produce significant improvements in working memory [10,11,12]. These studies indicated that improvements in short-term memory are positively correlated with an increase in the relative amplitude in the individual upper alpha band during training when the EEG signal is recorded from the Cz channel [11]. In another study, subjects with mild cognitive impairment were trained to increase the power of their individual upper alpha band of the electroencephalogram (EEG) signal over the central parietal region. An increase in peak alpha frequency was observed throughout the period of training, and the memory performance also improved significantly following the training [13]. Another NFT study provided promising results that demonstrate the trainability of frontoparietal alpha rhythm and its functional correlations with working memory and episodic memory [14]. These findings support the importance of alpha modulation in cognitive enhancements [5].

Although during NFT, a specific EEG activity over a specific brain region is used for training, this produces concomitant changes in brain regions and activities other than the trained one [12,15,16]. This indicates the interdependency of different oscillations of brain signals. The conventional taxonomy of brain oscillations is arbitrary, and brain signals are naturally non-isolated in-term of frequency ranges, which leads to this interdependency [17]. Besides the brain region and the frequency band, which is used during the training, investigating the effects of NFT in other brain regions and frequency bands can demonstrate the global underlying neural changes in the brain. The goal of the current study was, therefore, to analyse the data in the alpha band (8–13 Hz) (i.e., trained activity), as well as in the theta (4–8 Hz) and beta (13–30 Hz) bands (i.e., non-trained activities).

Cognitive activity recruits functional connectivity (FC) between regions distributed over the whole brain instead of a single brain region [18,19]. Along with the studies reporting the increase in spectral power, NFT modulates large-scale neural networks [20], and there is evidence of changes in the brain FC after NFT [17]. Such FC changes are observed in a task-based EEG activity after NFT [20]. The FC changes after NFT have been extensively studied in patients with neurological diseases, such as traumatic brain injuries [21] and autism [22]. However, the exploration of changes in FC after NFT in task-based EEG in healthy participants is lacking.

The processing of information in the left and right hemispheres of the human brain has unique properties that lead to functional asymmetry during memory tasks [23,24]. This asymmetric behaviour of the brain is known as hemispheric lateralization [25]. Different studies have identified a decrease in hemispheric lateralization associated with the cognitive demands during controlled inhibition [26], conflict resolution [27], and memory [28]. Asymmetric brain activation can also lead to differential FC in the two hemispheres. In general, the rationale behind this study is to explore the effect of alpha NFT on cognitive abilities. Specifically, in this paper, we want to see its effect on the working memory capacity in healthy participants by exploring the changes in the FC after NFT in an N-back memory task. Therefore, our first hypothesis is that the brain network in the left and right hemispheres are involved in different degrees during the N-back (1- and 2-back) task. Additionally, our second hypothesis is that FC in both hemispheres will be enhanced after NFT. For this reason, we explored the hemispheric lateralization of FC using an N-back working memory task in two different memory loads (1- and 2-back) [29] and evaluated the changes in FC before and after NFT, with the aim to assess our hypothesis. The n-back task is a continuous performance task to measure working memory capacity in a way complementary to other established working memory tasks [19], so that it provides evidence on facets of working memory not typically addressed by other well-established working memory tasks.

## 2. Materials and Methods

### 2.1. Participants

Initially, 50 healthy participants (18 females and 32 males) were recruited via email advertisement over the campus. The participants were equally divided into two groups, the NFT group and the control group, and the assignments in terms of number of females in each group, were roughly equal. They were university students. The mean age of the NFT group was 23.32 ± 4.66 years, and that of the control group was 26 ± 4.84 years, with no significant difference in the group ages. Due to unavoidable circumstances (e.g., their scheduled session clashed with their classes), some of the participants could not participate in the study until the end, and therefore, they were excluded from the analysis. Additionally, the data of some the participants were also excluded from the analysis due to the excessive artifacts in their N-back EEG recordings. The final sample size for the analysis in the current study included the data from 36 participants (25.45 ± 5.36 years old; 12 females and 24 males). Before conducting the experiments, the researcher conducted a one-to-one interview with each participant and verbally asked about their health (e.g., if they currently have any health issues or had any psychological problem in the past). All participants were healthy and had no psychological problems. Participants were randomly assigned to either the NFT group (17 participants) or the control group (19 participants). They voluntarily participated in the study, and no remuneration was provided. All experimental procedures were evaluated and approved by the scientific and ethical review committee of the Institute of Post-graduate Studies and Research (IPSR), Universiti Tunku Abdul Rahman, Malaysia (ref. number U/SERC/90/2018) and were in accordance with the Declaration of Helsinki. Before the start of the experiments, each participant was briefed by the researcher in a one-on-one meeting about the purpose and procedures of the study. A written informed consent form was signed by each participant.

### 2.2. Neurofeedback Training

The NFT was performed using Openvibe [30], which is an open source BCI software toolbox. A customized NFT protocol was designed. An Emotiv Epoc+ device was used for EEG signal acquisition; it consists of 14 active EEG sensors, which were placed on the scalp at the AF3, F7, F3, FC5, T7, P7, O1, O2, P8, T8, FC6, F4, F8, and AF4 locations according to 10–20 international standards. Despite the fact that the placement of the sensors from this device are not customizable, it has been used in many studies and has proved to be an effective wireless and low-cost device for NFT research [31].

The frontal lobe of the brain plays an important role in the higher level of cognitive functions and is responsible for immediate and sustained attention, time management, social skills, emotions, empathy, working memory, and executive planning [32]. Furthermore, the alpha band is the most prevalent rhythm in adult EEG and related to psychological states and cognitive processing [8]. The frontal alpha activity caused by thalamic and anterior cingulate cortex activity addresses attention and working memory [33]. Therefore, for neurofeedback, the Alpha EEG rhythm (8–13 Hz) from the frontal sensors F7, F3, F4, and F8 were processed in real time via the signal processing module designed in OpenVibe [34]. The Butterworth filter with 0.5 dB pass band ripples was used to extract the alpha band from the raw EEG signals through the “Modifiable Temporal Box” in OpenVibe. Welch’s method with a Hamming window was used to compute the absolute value of the alpha band power using a 5 seconds window [35]. Human psychophysics indicates an impairment in the visual perception and performance with unsuitable feedback latency [36]. Therefore, considering these factors and the visual ergonomics of healthy people, the feedback was updated every second, meaning that there was 80% overlap with the previous segment. Power was computed for each sensor separately and, then, averaged over the sensors to use as feedback. The feedback was displayed to the participants in the form of a vertical bar on a monitor as feedback. This vertical bar was designed in Unity3D [37], which is also an open source software toolbox. OpenVibe and Unity3D communicate with each other with the help of a Lab Streaming Layer (LSL) network protocol. Figure 1B shows the connections between OpenVibe, Unity 3D, and LSL and the flow of the signal from the EEG device to the final output of the feedback bar.

An Acer swift, 8 GB RAM, Windows 10 OS, 13.5-inch native display computer was used for data acquisition and processing. The participants viewed the vertical alpha power bar, which was updated in real time, on a 17-inch LCD monitor connected to the main computer. The participants were seated in a chair in front of the 17-inch display monitor at a comfortable distance during the NFT session. The participants had to look at the vertical bar, so they were asked to keep their eyes open and to minimize the blinking.

The NFT group participants received training for a total of 600 min, which was divided into 20 sessions, where each session had a duration of 30 min (20 × 30 min = 600 min). Each 30 min session was further divided into two sub-sessions, each of 15 min duration. A short break of 1–2 min was provided between the two sub-sessions. Each participant received a minimum of one and a maximum of three NFT sessions per week. The participants in the NFT group took around 10 weeks from the beginning of the first NFT session to complete 20 sessions. Therefore, there was approximately 10 weeks in the gap between the pre-NF and post-NF N-back task. The time slot for the training was kept consistent across all the sessions for the participants to minimize the influence of circadian rhythm [38]. The present study is not a clinical study and should not be interpreted as such. The study’s aims were to provide new evidence on working memory mechanisms, not on the efficacy of alpha NFT in enhancing working memory capacity. Thus, the participants in the control group did not perform any training. They performed the N-back task at the start (which counted as the pre-NF session) and at the end (which counted as the post-NF session). To compare the NFT effects in an unbiased manner in terms of timing, the participants in the control group were invited for a second attempt at the task approximately 10 weeks after the first attempt (i.e., pre-NF session). Therefore, the gap between the pre-NF and post-NF N-back tasks were approximately equal for both groups.

Before starting the NFT session, the researcher instructed the participants to increase the height of the red vertical bar showing alpha power in real time (i.e., feedback bar) by adopting their own strategy. The complete design of the NFT protocol can be seen in Figure 1.

### 2.3. Experimental Protocol

The current study employed an N-back memory task to assess working memory capacity with two memory load factors, 1-back (low memory load) and 2-back (high memory load). The structure of the N-back test is shown in Figure 2. Twelve English letters—B, D, H, J, K, M, N, P, R, T, V, and X—in the 1-back condition and thirteen letters—B, C, F, H, J, K, M, N, P, Q, S, V, and X—in the 2-back condition in capital font were presented to the participant on the screen as the test stimuli. For each stimulus, they were required to press a key (1 or 2) on the keyboard to respond to the stimulus. If the stimulus letter was a target, the participant needed to press 1, and if the stimulus letter was a non-target, the participant needed to press 2. The test consisted of two conditions, 1-back and 2-back, each containing 36 stimuli; 12 of these were targets, and the other 24 were non-targets. In the 1-back condition, if the letter was the same as the one presented just before, it was the target letter; otherwise, it was a non-target. Similarly, in the 2-back condition, a letter was a target if it was the same as the letter presented two before. Each stimulus appeared for a maximum of 1500 ms on the screen. The participant was required to respond to the stimulus within this time. Between the two consecutive stimuli, an inter stimulus interval (ISI) of 500 ms was presented in the form of a fixation (i.e., a plus sign).

Before starting the test, the researcher orally gave the instructions to the participant. The task initiated with the 1-back condition, and then, the 2-back condition followed. The sequence of test presentation was the same for all participants. Twenty practice trials preceded 36 test trials for each condition. When the practice trials were completed, the participant was informed via onscreen instructions that “practice trials are finished and now you will be presented with actual test trials”. The stimuli were presented on an extended display of 17-inch LCD via E-Prime version 3.0 (Psychology Software Tools, Pittsburgh, PA, USA) stimulus presentation software.

### 2.4. Task-Based Data

#### 2.4.1. EEG Recordings and Processing

The continuous EEG data during the task performance were recorded using an Emotiv Epoc+ EEG device containing 14 active EEG sensors located at AF3, F7, F3, FC5, T7, P7, O1, O2, P8, T8, FC6, F4, F8, and AF4 and two reference electrodes placed on the mastoid bones behind the ears. The sampling rate of the device was 128 samples/seconds. We combined the AF3, F7, F3, F4, F8, and AF4 sensors into the frontal lobes; FC5 and FC6 into the fronto-central lobes; P7 and P8 into the parietal lobes; and O1 and O2 into the occipital lobes to perform analyses on each lobe separately.

The recorded EEG signal was filtered offline using a 0.1 Hz high-pass FIR filter to remove the slow drift and a 40 Hz low-pass FIR filter via the *pop_eegfiltnew* function in the EEGLAB toolbox [39] with a Hamming window and default options of the transition bandwidth. The narrowband EEG data were then passed through multiple data cleaning stages to remove the artifacts from the data. Artifact Subspace Reconstruction (ASR) is a widely used method for cleaning EEG data and uses the variance-based algorithm presented in [40]. This algorithm creates a statistical model of the clean EEG portion in the data and applies principal component analysis (PCA) to new incoming raw signal and transforms it into the principal component (PC) space. If any of the PCs have larger variances than the variance of the calibration data, it is rejected, and the signal is reconstructed and projected back into the original channel data [41]. This method is implanted with the *pop_cleanrawdata* function in the EEGLAB. We cleaned the data with ASR and, then, segmented it into short epochs from −500 to 1000 ms around the stimulus onset time (i.e., 0 ms), resulting in a 1500 ms epoch length. The EEG segments of correct-only trials were included in further processing and the final analysis [42]. Each EEG segment was inspected visually, and every segment containing artifacts was rejected. Secondly, the independent component analysis (ICA) method using the infomax algorithm [43] was applied to each segment of the data. ICA is a signal processing tool that separates linearly mixed sources. In EEG recordings, these sources are the brain signals (i.e., true EEG) plus some common physiological artifacts, such as an eye artifact or EMG. These artifacts are not the true EEG signals and are referred to as non-brain activity. An eye artifact has an almost entirely frontal distribution over the scalp, and ICA has shown great success in identifying these artifacts in EEG data. Using ICA, the data were transformed into the component time-series space. A thorough visual inspection of each independent component (IC) was undertaken within EEGLAB by simultaneously looking at the time-series of that IC, its time–frequency plot, and its topographical map. The ICs that had mostly a frontal distribution, sudden peaks at the frontal regions (i.e., Fp1 and Fp2 electrodes), and characteristics that looked like non-brain activity were identified as eye artifacts and removed from the data. The signal was reconstructed with the remaining ICs. To avoid the chances of losing excess real EEG data, the maximum number of ICs to reject per recording was restricted to 3 out of 14, meaning that the number of ICs rejected per recording was between 1 and 3. The average number of trials per subject included in the final analysis for the NFT group in 1-back were 30.3 and 28.8 and in 2-back were 26.5 and 27.8 in the pre-NF and post-NF sessions, respectively. Similarly, in the control group, the average trials in 1-back were 31.5 and 32.1 and those in 2-back were 26.1 and 29.6 in the pre-NF and post-NF sessions, respectively.

The Laplacian filter is a spatial high-pass filter that effectively attenuates the low spatial frequencies that can potentially cause volume conduction artifacts in the sensor-level EEG signals. Thus, applying the Laplacian filter is appropriate for EEG data before the connectivity analysis [44]. Filtering the data using a Laplacian filter enhances the signal-to-noise ratio and has proved effective in the EEG data, e.g., in brain–computer interfaces [45], it reasonably suppresses the volume conduction artifact in the data and makes it appropriate for the connectivity analysis [46]. Therefore, we also applied the Laplacian filter as a pre-processing step to attenuate the volume conduction artifacts before the functional connectivity (FC) analysis. We implemented the Laplacian filter using the spherical splines approach, and the filter has an order of 2, as described in [47].

#### 2.4.2. Time–Frequency Decomposition

The EEG signal is highly non-stationary in nature [35]. To overcome the non-stationarity characteristics of the EEG data, the signal is decomposed into short time windows, which are considered roughly stationary via time–frequency decomposition [48]. Complex Morlet wavelet (CMW) convolution is one of the time–frequency decomposition methods that results in optimal trade-off between the time precision and spectral resolution, which is the main factor in a time–frequency analysis [48]. Therefore, we also used the CMW convolution method for time–frequency decomposition. CMW is defined by multiplying a complex sine wave with the Gaussian wave, as given in Equation (1);
(1)$$cmw=e^{i2\pi ft}e^{\left(-t^{2}/2\sigma^{2}\right)}$$
where *t* is the time and *f* is the frequency. In the current analysis, the frequency is defined from 1 to 30 Hz using 40 logarithmic steps. σ is the Full-Width at Half-Maximum (FWHM) parameter and is given by Equation (2);
(2)$$FWHM=\frac{n}{2\pi f}$$

In Equation (2), n represents the number of wavelet cycles, which is fixed from 3 to 8 in 40 logarithmic steps in the current study.

#### 2.4.3. Power Computation

The time–frequency decomposition of EEG trials resulted in a complex output. The post-stimulus instantaneous power for each trial was computed by taking the square of the real and imaginary parts of the output [17]. Each trial was baseline-corrected using a pre-stimulus sub-window from −300 ms to −100 ms (where 0 ms is stimulus onset). For power-law compensation, a baseline normalization was performed, and the values were converted to the decibel scale (i.e., decibel transformation) [49] using Equation (3).
(3)$$Power\;\left(in\;dB\right)=10\left(\frac{power}{baseline}\right)$$
where the baseline activity was taken between −300 ms and −100 ms. Finally, the trials were averaged to compute the mean power.

#### 2.4.4. Functional Connectivity Measurement

To identify individual rhythmic components that compose the measured data, specifically to study rhythmic neuronal interactions, frequency domain representation of the signal is often considered convenient [50]. Subsequently, frequency-domain connectivity metrics can be estimated to evaluate the neuronal interactions. These matrices include the quantification of consistency across observations of the phase difference between the oscillatory components in the signals, for example, phase locking value (PLV). The distribution of the phase differences computed via the PLV method could be indicative of functionally meaningful synchronization between neural populations. Therefore, we preferred to use this method in the current analysis.

PLV is a phase synchronization-based measure for FC analysis in BCI research [51] to measure the FC between different EEG electrodes. PLV represents the average phase difference between two electrodes, ranges between 0 (no synchronization) and 1 (full synchronization), and is given by Equation (4).
(4)$$PLV\left(t\right)=\frac{1}{N}\left|\overset{N}{\underset{n=1}{\sum }}exp\left(\;j\left(\Delta \varphi_{n}\left(t\right)\right)\;\right)\right|$$
where $\Delta \varphi_{n}\left(t\right)=\varphi_{x}\left(t\right)-\varphi_{y}\left(t\right)$ represents the phase difference between electrode x and electrode y, t is time, and N is the length of the time-series. The broadband signal was filtered in the desired EEG band, and the analytic signal was computed. Fast Fourier transform (FFT) and Hilbert transform are two prominent methods to compute the analytic signal. We chose to use Hilbert transform because it is very useful in analysing nonstationary signals such as EEG [52]. Then, the time series of the phase angles from the analytic signal were obtained for each EEG electrode. Finally, PLV was computed for each epoch using Equation (4) and, then, averaged across epochs. The PLV values were computed for each epoch; therefore, the window length of the PLV calculation was equal to the length of the epoch (i.e., 1500 ms). The brain networks were visualized using BrainNet Viewer [53], a MATLAB toolbox for brain connectivity analysis.

### 2.5. EEG Data Collected during the NFT Sessions

The pre-processing for the EEG-data collected during the NFT sessions is described in detail in our publication [34] and is presented here for completeness. The pre-processing was partly similar to the pre-processing of task-based EEG data. First, continuous EEG data were filtered via a FIR filter between 0.5 and 40 Hz using a Hamming window with the help of the pop_eegfiltnew function in EEGLAB. After that, ASR was used to reconstruct the artifact portion of the data with clean data [40]. This method removes the non-stationary high-variance signal and, then, reconstructs the missing data using a spatial mixing matrix. Then, the data were segmented into 2 s short epochs, and each epoch was inspected for the artifact using the pop_autorej function in EEGLAB. An epoch was selected as a bad epoch if any of the data points exceeded five standard deviations of the amplitude and was rejected from the data. This algorithm was iteratively implemented via a custom-written MATLAB script. If the number of epochs selected for rejection was greater than 5% of the data, the procedure was repeated with a more liberal threshold (increased by 0.5 SD) [54].

Absolute alpha power was computed using Welch’s averaged, modified periodogram method (512 DFT points and 75% overlap between consecutive windows) and a Hann tapering window. The power was computed for each channel separately, and then, the mean power was computed across all 14 EEG channels.

### 2.6. Statistical Analysis

In the current study, we have two groups (NFT and control) and two sessions (Pre-NF and Post-NF). Therefore, a 2(Group: NFT vs. Control) × 2(Session: Pre-NF vs. Post-NF) two-way repeated measure analysis of variance (ANOVA) was performed. In the ANOVA model, the between-subject factor was Group and the within-subject factor was Session. The same ANOVA model was applied separately for the 1-back and 2-back conditions and were repeated for behavioural results (i.e., RT and error rates), band powers, and FC. A paired t-test with Bonferroni correction was performed for pairwise comparisons where needed [55]. All statistics were performed using the Pingouin statistical package implemented in Python [56].

The EEG data collected during the NFT sessions were analysed to investigate whether the within-session alpha power changed in response to the training. For this purpose, we compared the mean alpha power within the 1st half (i.e., first 15 min) and 2nd half (i.e., last 15 min) of the NFT session to the baseline using separate one-way repeated measure ANOVA. The EEG data collected before starting the experiments were used as the baseline in this case. Additionally, we explored the within-session changes by investigating each half of the NFT session. We divided each half of the NFT session into three five-minute sub-blocks (B1, B2, and B3). Then, we applied a 2 (half session: 1st half and 2nd half) × 3(sub-blocks: B1, B2 and B3) 2-way ANOVA to investigate the alpha power differences between the 1st half and 2nd half and among B1, B2, and B3.

## 3. Results

### 3.1. Behavioural Results (Response Time and Error Rate)

The behavioural results are shown in Figure 3. In the response times (RT), there was an expected significant main effect of Session in the 1-back condition, $F\left(1,34\right)=5.25,\;p=0.028,\;\eta^{2}=0.133$. RT significantly reduced from 549.2 (ms) to 544.69 (ms) in the Post-NF Session for the NFT group and from 604.27 (ms) to 533.89 (ms) in the control group. Similarly, in the 2-back condition, there was a significant main effect of Session, $F\left(1,34\right)=5.26,\;p=0.000016,\;\eta^{2}=0.426$, as well as an interaction effect between Group and Session, $F\left(1,34\right)=4.48,\;p=0.041,\;\eta^{2}=0.116$. RT in the Post-NF Session significantly reduced from 740.23 (ms) to 684.80 (ms) in the NFT group and from 759.25 (ms) to 619.40 (ms) in the control group.

In the 1-back condition, there was a significant main effect of Session on the error rates, $F\left(1,34\right)=5.98,\;p=0.019,\;\eta^{2}=0.149$. The error rates in the Post-NF Session decreased to 2.2% from 4.9% in the NFT group and to 4.9% from 7.3% in the control group. Similarly, the main effect of the Group was also significant, $F\left(1,34\right)=8.21,\;p=0.007,\;\eta^{2}=0.194$. However, there was no interaction effect between Group and Session, $F\left(1,34\right)=0.01,\;p=0.892,\;\eta^{2}=0.0005$. In the 2-back condition, the main effects of Session and Group were significant $F\left(1,34\right)=15.70,\;p=0.0003,\;\eta^{2}=0.315$ and $F\left(1,34\right)=4.34,\;p=0.044,\;\eta^{2}=0.113$, respectively. The error rates in the Post-NF session decreased to 9.34% from 13.5% in the NFT group and to 12.42% from 20.17% in the control group. However, the interaction effect between Group and Session was not significant $F\left(1,34\right)=1.26,\;p=0.26,\;\eta^{2}=0.036$.

### 3.2. Within-Session Alpha Power

The within-session mean alpha power for the 1st half and 2nd half of the NFT session is shown in Figure 4. One-way repeated measure ANOVA revealed that the within-session alpha power was significantly greater than the baseline alpha power (*p* = 0.00207). The post hoc test revealed that the alpha power in both halves of the session was significantly greater than that of the baseline, with *p* = 0.0255 and *p* = 0.046 for the first half and second half, respectively. However, the alpha power in the first half and second half was not significantly different from each other (*p* = 0.25956).

Overall, the alpha power within each half of the session significantly increased from the baseline. However, this increase reduced from the beginning (B1) to the end (B3) of the session, as shown in Figure 4. From the 2-way ANOVA, we found that this decrease (from B1 to B3) within each half of the session was significant (*p* = 0.00076). The post hoc test revealed that this significant main effect was observed between B1 and B3 (*p* = 0.0060) and between B2 and B3 (*p* = 0.0183). However, the difference between B1 and B2 was not significant (*p* = 0.22516).

### 3.3. EEG Post-Stimulus Power

#### 3.3.1. Frontal Lobe

In 1-back condition, there was no significant main effect of Group or Session in frontal theta, alpha, and beta activities. Similarly, in the 2-back condition, no significant main effect of Group or Session was found in frontal beta activity. The main effect of Group in frontal alpha activity and the main effect of Session in theta activity were also non-significant. However, a significant main effect of Group in frontal theta, $F\left(1,34\right)=9.01,\;p=0.0049,\;\eta^{2}=0.20$, and a marginally significant effect of Session in frontal alpha, $F\left(1,34\right)=3.91,\;p=0.055,\;\eta^{2}=0.10$, were observed. The post-stimulus power for the frontal lobe is shown in Figure 5.

#### 3.3.2. Fronto-Central Lobe

In the 1-back condition, the main effect of both factors, Group and Session, were non-significant in alpha and beta activities. Similarly, the main effect of Session was also non-significant in theta activity. However, the main effect of Group was significant in theta activity, $F\left(1,34\right)=5.42,\;p=0.025,\;\eta^{2}=0.137$. In the 2-back condition, there was no significant main effect of Group or Session in beta activity. On the other hand, in theta activity, the main effect of Group, $F\left(1,34\right)=9.79,\;p=0.0035,\;\eta^{2}=0.223,\;$ as well as of Session, $F\left(1,34\right)=7.22,\;p=0.011,\;\eta^{2}=0.175$, were significant. Similarly, in alpha activity, the main effect of both factors, Group, $F\left(1,34\right)=4.55,\;p=0.040,\;\eta^{2}=0.118,\;$ and Session, $F\left(1,34\right)=8.18,\;p=0.0071,\;\eta^{2}=0.194$, were significant. The fronto-central lobe post-stimulus power results are shown in Figure 6.

#### 3.3.3. Parietal Lobe

In the 1-back condition, the main effect of Group was significant in theta, $F\left(1,34\right)=8.19,\;p=0.0071,\;\eta^{2}=0.194$, and alpha activities, $F\left(1,34\right)=10.34,\;p=0.0028,\;\eta^{2}=0.233$, and non-significant in beta activity. The main effect of Session was non-significant in theta and alpha activities. However, there was a significant effect of Session in beta activity,$\;F\left(1,34\right)=4.80,\;p=0.0352,\;\eta^{2}=0.123$. In the 2-back condition, no significant main effects for Group or Session, were observed in theta, alpha, or beta activities. However, the main effect of Group was significant in theta activity,$\;F\left(1,34\right)=8.71,\;p=0.0056,\;\eta^{2}=0.204$. The post-stimulus power results for the parietal lobe are shown in Figure 7.

#### 3.3.4. Occipital Lobe

In the 1-back condition, the interaction effect of Group and Session was significant in theta activity, $F\left(1,34\right)=4.55,\;p=0.0402,\;\eta^{2}=0.118$. The main effect of Group was significant in theta, $F\left(1,34\right)=12.89,\;p=0.0010,\;\eta^{2}=0.275$; alpha, $F\left(1,34\right)=13.58,\;p=0.00079,\;\eta^{2}=0.285$; and beta activities, $F\left(1,34\right)=6.71,\;p=0.0139,\;\eta^{2}=0.165$, whereas the main effect of Session was non-significant. In the 2-back condition, the main effect of Group was significant only in theta activity,$\;F\left(1,34\right)=9.08,\;p=0.0048,\;\eta^{2}=0.210$. The main effect of Session was significant in theta, $F\left(1,34\right)=5.56,\;p=0.0241,\;\eta^{2}=0.140$, and beta activities, $F\left(1,34\right)=4.20,\;p=0.0480,\;\eta^{2}=0.110$. No significant interaction effect was observed in the 2-back condition. The post-stimulus power results for occipital lobe are shown in Figure 8.

### 3.4. EEG Functional Connectivity

#### 3.4.1. Theta Band

The connectivity results for theta band are shown in Figure 9. There was no significant main effect of Group or Session on the FC in the frontal lobe. However, the interaction effect between Group and Session was significant in the 1-back condition, $F\left(1,34\right)=9.538,\;p=0.0040,\;\eta^{2}=0.219$, as well as in the 2-back condition, $F\left(1,34\right)=6.68,\;p=0.0142,\;\eta^{2}=0.164$. A post hoc test with Bonferroni correction revealed that the FC in the frontal lobe of NFT group in both experimental conditions, 1-back and 2-back, significantly increased in the Post-NF session when compared with the Pre-NF session (*p* < 0.05).

The main effect of Group was also non-significant in the fronto-central lobe. However, the main effect of Session (1-back: $F\left(1,34\right)=5.85,\;p=0.0210,\;\eta^{2}=0.146$, 2-back: $F\left(1,34\right)=4.96,\;p=0.0326,\;\eta^{2}=0.127$) and the interaction effect between Group and Session (1-back: $F\left(1,34\right)=6.25,\;p=0.0174,\;\eta^{2}=0.155$, 2-back: $F\left(1,34\right)=5.17,\;p=0.0293,\;\eta^{2}=0.132$) were significant. The post hoc test again revealed a significant increase in FC of the NFT group in both conditions, 1-back and 2-back, in the Post-NF session when compared to the Pre-NF session (*p* < 0.05).

Similarly, in the parietal lobe, the main effect of Group was non-significant and the main effect of Session, $F\left(1,34\right)=4.29,\;p=0.0460,\;\eta^{2}=0.112$, as well as the interaction effect, $F\left(1,34\right)=9.83,\;p=0.0035,\;\eta^{2}=0.224$, were significant in the 1-back condition. In the 2-back condition, the main effect of Session was non-significant, but the interaction effect was significant, $F\left(1,34\right)=8.47,\;p=0.0063,\;\eta^{2}=0.199$. The post hoc test revealed a significant increase in Post-NF FC in the NFT group only (*p* < 0.05).

Finally, in the occipital lobe, the main effect of Group and Session was non-significant in both conditions, 1-back and 2-back. However, the interaction effect between Group and Session was significant in the 1-back condition, $F\left(1,34\right)=11.31,\;p=0.0019,\;\eta^{2}=0.249$, as well as in the 2-back condition, $F\left(1,34\right)=10.22,\;p=0.0030,\;\eta^{2}=0.231$. Similarly, to other brain regions, the post hoc test revealed a significant increase in Post-NF FC in the NFT group only (*p* < 0.05).

#### 3.4.2. Alpha Band

The results of the alpha band connectivity are shown in Figure 10. There was no main effect of Group or Session or an interaction effect between Group and Session in the 1-back condition in the frontal or fronto-central lobes. Similarly, in the 2-back condition, the main effect of Session and the interaction effect were non-significant in the fronto-central lobe. However, the main effect of Group in the 2-back condition in the fronto-central lobe was significant,$\;F\left(1,34\right)=4.58,\;p=0.0395,\;\eta^{2}=0.11$8.

The main effect of Group and Session and the interaction effect were also non-significant in the parietal lobe in both conditions, 1-back and 2-back. Similarly, in the occipital lobe, the interaction effect and main effect of Session were non-significant in the 1-back condition. However, the main effect of Group was significant, $F\left(1,34\right)=4.19,\;p=0.0483,\;\eta^{2}=0.109$, in the occipital lobe in the 1-back condition. No main or interaction effect of Group and Session was observed in the occipital lobe in the 2-back condition.

#### 3.4.3. Beta Band

The beta band connectivity results are shown in Figure 11. There was no main or interaction effect of Group and Session in the frontal lobe in either condition, 1-back or 2-back. In the fronto-central lobe, the main effect of Group was significant in the 1-back condition, $F\left(1,34\right)=7.138,\;p=0.0114,\;\eta^{2}=0.173$, as well as in the 2-back condition, $F\left(1,34\right)=5.65,\;p=0.0231,\;\eta^{2}=0.142$. However, the main effect of Session and the interaction effect were non-significant in the fronto-central lobe in both conditions, 1-back and 2-back. In the parietal and occipital lobes, no main or interaction effects were observed in either condition.

## 4. Discussion

The rationale behind the current study was to investigate the effects of alpha NFT on the alpha as well as theta and beta bands. The real-time feedback activity was recorded at the frontal lobe, and the offline analyses were extended to the frontal, fronto-central, parietal, and occipital brain regions. Additionally, based on our hypothesis, we visually observed the hemispheric lateralization of FC using the N-back task and found an enhancement in the brain FC in the theta band after NFT. Regarding the effect of NFT on the brain alpha, theta, and beta activities in the N-back task, we found different results in the post-stimulus power of both groups.

At the frontal region, before and after NFT, the post-stimulus alpha power presented similar levels in the low-memory condition (i.e., 1-back) and increased power in the high-memory condition (2-back). The increased activation in the 2-back condition for both groups in the post-NF session is suggestive of the involvement of the frontal region in the more-demanding high-memory condition [57]. From the pre- to post-NF sessions in the frontal lobe, a similar change in both groups was observed, which indicated a non-specific activation that cannot be attributed to NFT effects. Previous research stated that a more circumscribed cortical activation is related to better working memory performance [58]. A load-dependent change in theta and alpha band at the fronto-central, parietal, and occipital positions was observed with increased theta and alpha activities in the high-load condition (i.e., 2-back task). Regarding alpha activity, a similar effect was observed, where an increase in alpha activity with increasing memory load was observed [59]. Similarly, research studies have shown an increase in theta EEG power with the increase in working memory load [60,61,62]. The power changes observed in different EEG bands in the current study contribute to an understanding of the neural mechanism of working memory. Along with the increase in training band (i.e., alpha), enhanced activity in the non-training band (i.e., theta) was also observed. This increase was not really a result of the NFT but it represents a strong coordination between alpha and theta activities during a memory task per se.

A visual observation of FC across the theta, alpha, and beta bands revealed hemispheric lateralization. Our statistical analysis confirmed that only the theta band FC was correlated with working memory and was increased after training. Previous research associated the theta band in a human being with working memory [63], and a significantly lower FC in the theta band was reported in memory impairments [64]. The integration of a brain process to confront the increase in memory demands during a working memory task was indicated only in the increase in theta FC [65]. A higher FC in the more efficient group within the theta band was observed [66], indicating that their working memory function seems to be more efficient in contrast with others. Sauseng et al. proposed a ‘process level’ mechanism for theta synchronization during the encoding phase of working memory, which integrates processes such as attention across brain regions [67]. These brain processes are very crucial in the formation of a memory trace. In the current study, the enhanced FC in the theta band suggests the presence of stronger interregional synchronization in the NFT group when compared with the control group: an important ingredient for the co-activation of neural structures, which is involved in different sub-processes of working memory function [67]. These results suggest that upper alpha training elicited reorganization processes in the brain, which are not only predominant in the right hemisphere but also visible in a different frequency band, the theta band. Research has indicated that working memory deficits are associated with lower connectivity of the dorsolateral prefrontal cortex (DLPFC) with other regions, and the higher global brain connectivity in our results within the NFT group predicts better working memory performance as well as general fluid intelligence [68]. Interactions between the alpha and theta bands have also been reported [69]. Both alpha and theta rhythms have been postulated to mediate the interaction between distal and widely distributed connections. Since the alpha rhythm is distributed across the whole scalp, it is highly suitable to engender neural plasticity and, consequently, adaptive changes in the brain activity that can be seen foremost in the theta frequency. The increase in theta band power has been associated with increased encoding success in memory tasks [8]. Theta oscillations have been related to control processes and regional integration during working memory [67]. These results indicate that FC strength in the theta band was sensitive to working memory capacity and was aided by neurofeedback. A generally lower FC is associated with a lower performance in several cognitive domains and is mostly observed empirically in patients, e.g., those with schizophrenia [70]. The results from the present study particularly highlight a segregation of theta FC between the left and right hemispheres when assessing working memory capacity. Furthermore, neurofeedback seems to play a major role in manipulating the theta network of working memory capacity, which is in line with our hypotheses.

Previous research has demonstrated that FC in the resting condition is inversely related to cognitive reserve [71], but there is also evidence of a higher FC in the theta band, which suggests higher cognitive reserve scores for active engagement and attention during a task [72]. The relevance of theta activity for attention in humans has been reported in [73], and increased global connectivity over the brain in theta band has been reported to enhance cognitive performance. Finally, we conclude that the increased FC in the theta band in the NFT group after training in the current study reflects the positive effect of training on continuous cognitive processing (attention and executive control) during task performance [74]. As an indicator of higher cognitive reserve scores [75], the theta FC patterns in our results reflect the more efficient functional networks of attention and executive function during memory processing after the training [74].

The following limitations of the present study need to be addressed in future studies: A gender-balanced design is necessary to understand the extent and generality of lateralization effects. Since brain lateralization is not comparable in male and female individuals, a sample with a balance of genders should help to provide a more precise generalization of the study results [76]. Moreover, the addition of a second control group receiving the placebo NFT would extend the present results to the realm of clinical usability.

## 5. Conclusions

In our results, we found that alpha neurofeedback training might improve the working memory capacity of healthy participants, as evidenced by the enhanced functional brain connections in the theta band. From the visual inspection of the plots of the brain networks, we observed that the functional connectivity of working memory within each hemisphere of the brain is stronger in comparison with the connectivity between the two hemispheres.

## Figures and Tables

**Figure 1 bioengineering-10-00200-f001:**
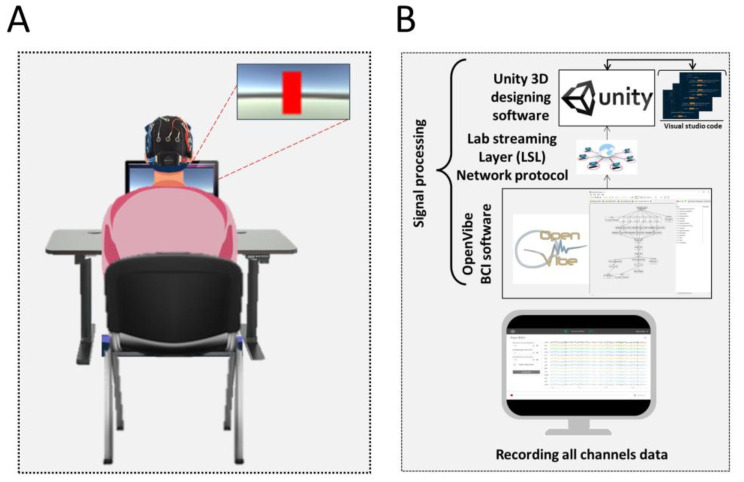
Representation of the NFT training system. (**A**) A participant is sitting in front of the feedback screen, and the visual feedback is shown as a red vertical bar on the screen. (**B**) The EEG data are recorded for the offline analysis, and the signal processing module in OpenVibe is connected with the Unity3D via LSL protocol.

**Figure 2 bioengineering-10-00200-f002:**
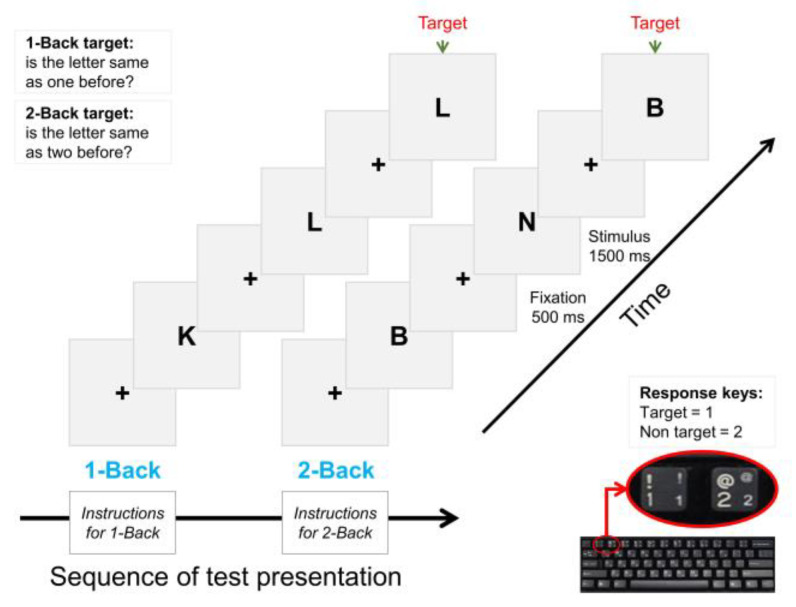
Structure of the N-back task.

**Figure 3 bioengineering-10-00200-f003:**
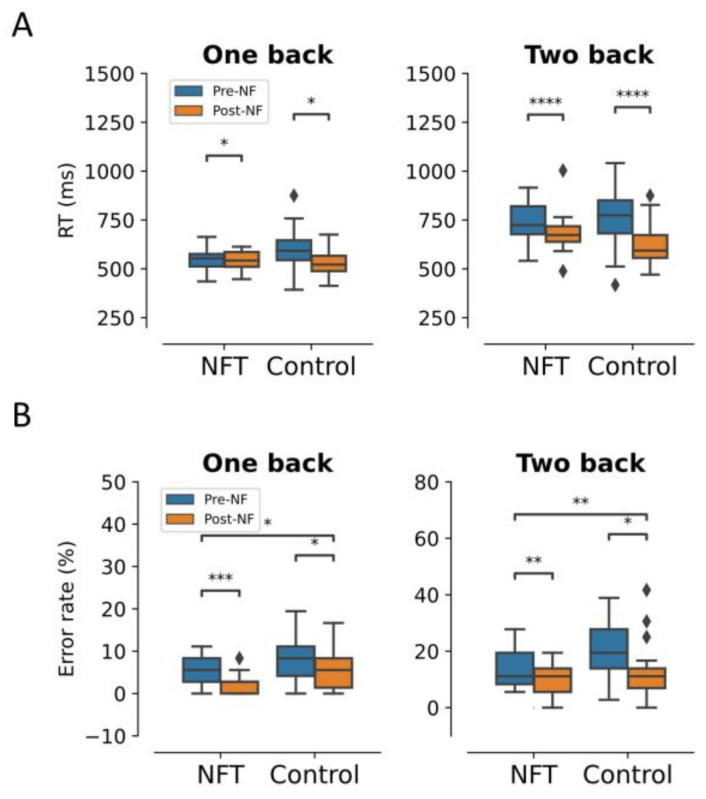
Behavioural results: (**A**) response time and (**B**) error rates. (* *p* < 0.05, ** *p* < 0.005, *** *p* < 0.0005, **** *p* < 0.00005). The diamond shape in the box plots represents the outliers in the data.

**Figure 4 bioengineering-10-00200-f004:**
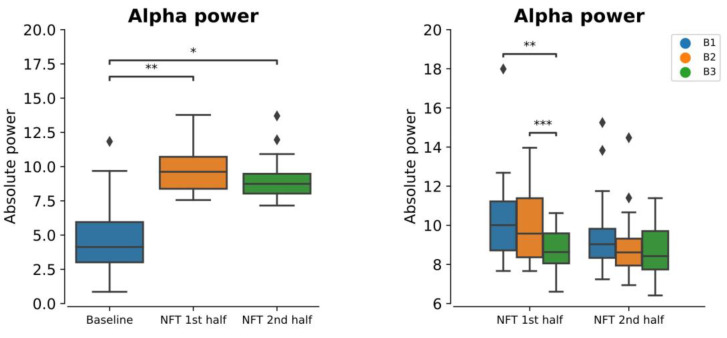
Absolute alpha power at baseline and within-session (1st half = first 15 min, 2nd half = last 15 min) and within-session (first half and second half) sub-blocks (B1, B2, and B3, each five minutes) (* *p* < 0.05, ** *p* < 0.005, *** *p* < 0.0005). The diamond shape in the box plots represents the outliers in the data.

**Figure 5 bioengineering-10-00200-f005:**
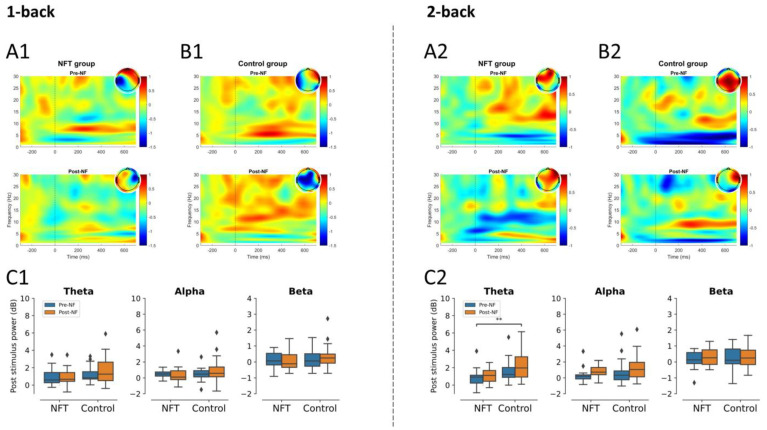
Post-stimulus power in frontal lobe (** *p* < 0.005): 1-back post-stimulus power in (**A1**) NFT group and (**B1**) control group; 2-back post-stimulus power in (**A2**) NFT group and (**B2**) control group. Post-stimulus power in different EEG bands within each group in (**C1**) 1-back and (**C2**) 2-back conditions. The colour bar of the TF plots represents the value of EEG power in dB scale. The diamond shape in the box plots represents the outliers in the data.

**Figure 6 bioengineering-10-00200-f006:**
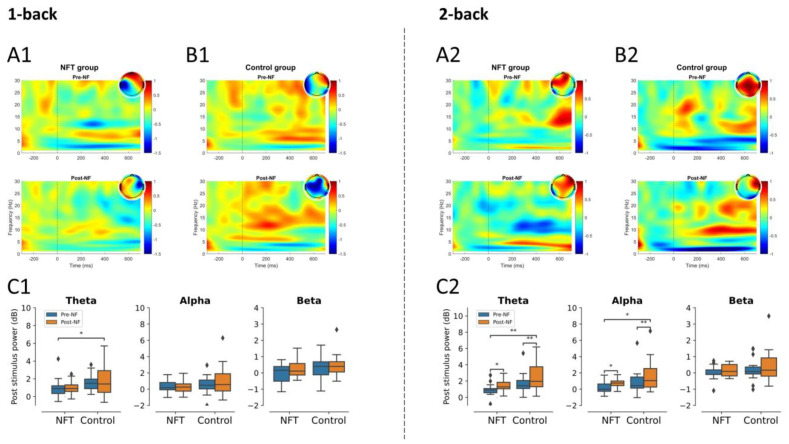
Post-stimulus power fronto-central lobe (* *p* < 0.05, ** *p* < 0.005): 1-back post-stimulus power in (**A1**) NFT group and (**B1**) control group; 2-back post-stimulus power in (**A2**) NFT group and (**B2**) control group. Post-stimulus power in different EEG bands within each group in (**C1**) 1-back and (**C2**) 2-back conditions. The colour bar of the TF plots represents the value of EEG power in dB scale. The diamond shape in the box plots represents the outliers in the data.

**Figure 7 bioengineering-10-00200-f007:**
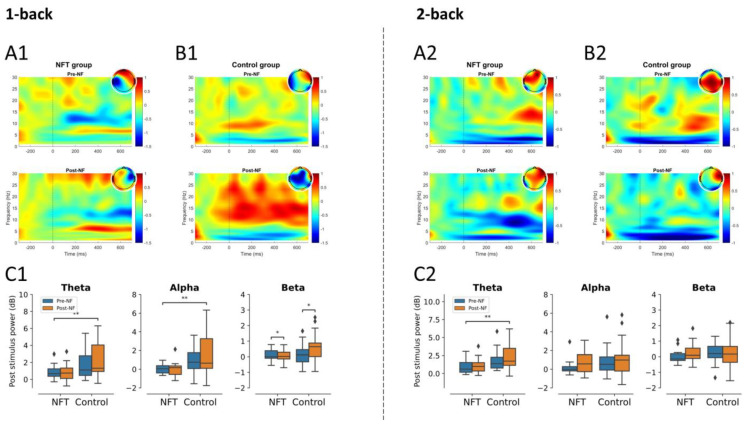
Post-stimulus power parietal lobe (* *p* < 0.05, ** *p* < 0.005): 1-back post-stimulus power in (**A1**) NFT group and (**B1**) control group; 2-back post-stimulus power in (**A2**) NFT group and (**B2**) control group. Post-stimulus power in different EEG bands within each group in (**C1**) 1-back and (**C2**) 2-back conditions. The colour bar of the TF plots represents the value of EEG power in dB scale. The diamond shape in the box plots represents the outliers in the data.

**Figure 8 bioengineering-10-00200-f008:**
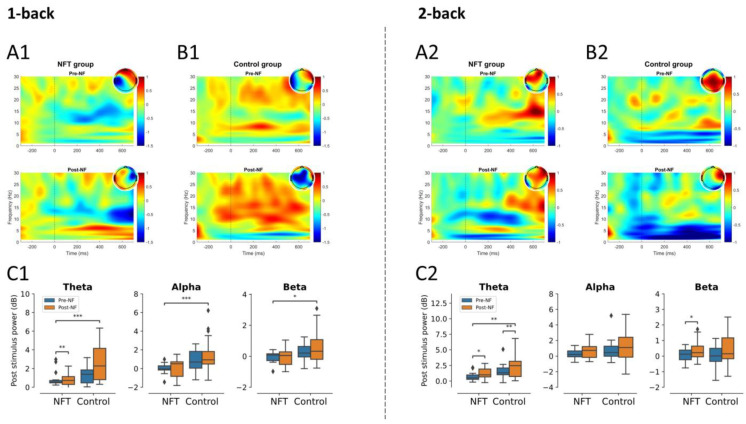
Post-stimulus power occipital lobe (* *p* < 0.05, ** *p* < 0.005, *** *p* < 0.0005): 1-back post-stimulus power in (**A1**) NFT group and (**B1**) control group; 2-back post-stimulus power in (**A2**) NFT group and (**B2**) control group. Post-stimulus power in different EEG bands within each group in (**C1**) 1-back and (**C2**) 2-back conditions. The colour bar of the TF plots represents the value of EEG power in dB scale. The diamond shape in the box plots represents the outliers in the data.

**Figure 9 bioengineering-10-00200-f009:**
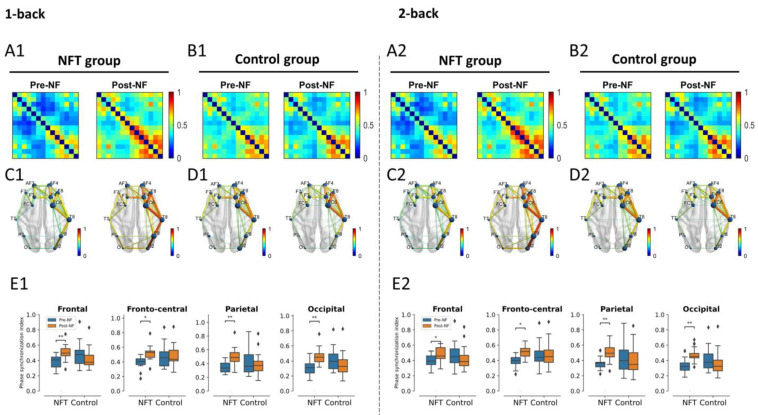
Theta band functional connectivity (* *p* < 0.05, ** *p* < 0.005). Connectivity matrices shown in (**A1**) for NFT group and (**B1**) for control group and brain connections in (**C1**) for NFT group and (**D1**) for control group in 1-back condition. Similarly, connectivity matrices shown in (**A2**) for NFT group and (**B2**) for control group and brain connections in (**C2**) for NFT group and (**D2**) for control group in 2-back condition. The box plots in (**E1**,**E2**) show the phase synchronization index in different brain lobes in 1-back and 2-back conditions, respectively. The colour bar of the connectivity matrices and brain connections represent the strength of synchronization between the EEG channels. It ranges from 0 (no synchronization) to 1 (full synchronization). The diamond shape in the box plots represents the outliers in the data.

**Figure 10 bioengineering-10-00200-f010:**
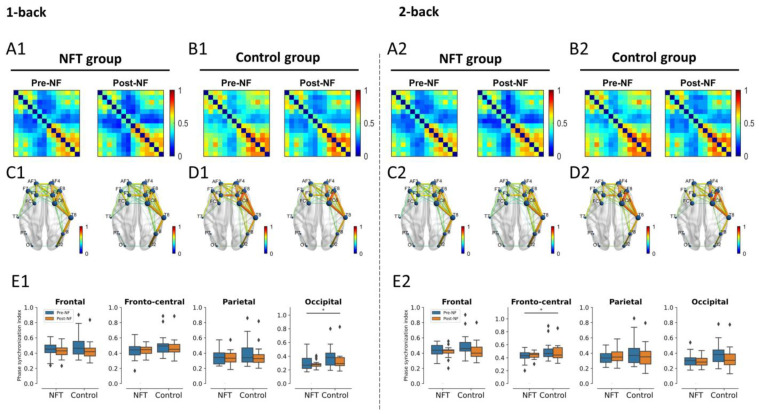
Alpha band functional connectivity (* *p* < 0.05). Connectivity matrices shown in (**A1**) for NFT group and (**B1**) for control group and brain connections in (**C1**) for NFT group and (**D1**) for control group in 1-back condition. Similarly, connectivity matrices shown in (**A2**) for NFT group and (**B2**) for control group and brain connections in (**C2**) for NFT group and (**D2**) for control group in 2-back condition. The box plots in (**E1**,**E2**) show the phase synchronization index in different brain lobes in 1-back and 2-back conditions, respectively. The colour bars of the connectivity matrices and brain connections represent the strength of synchronization between the EEG channels. It ranges from 0 (no synchronization) to 1 (full synchronization). The diamond shape in the box plots represents the outliers in the data.

**Figure 11 bioengineering-10-00200-f011:**
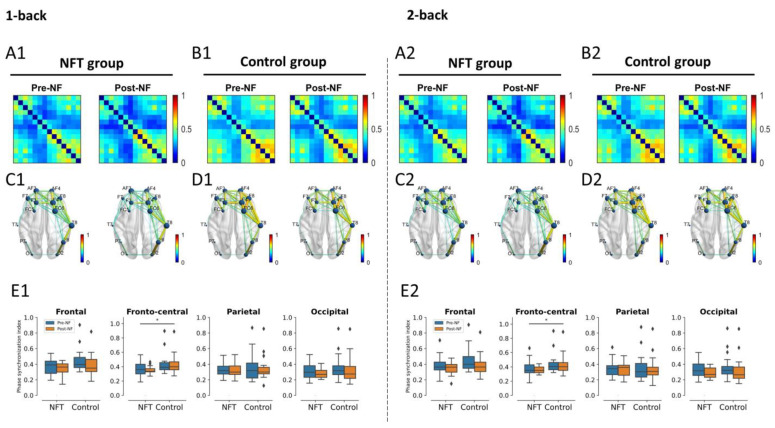
Beta band functional connectivity (* *p* < 0.05). Connectivity matrices shown in (**A1**) for NFT group and (**B1**) for control group and brain connections in (**C1**) for NFT group and (**D1**) for control group in 1-back condition. Similarly, connectivity matrices shown in (**A2**) for NFT group and (**B2**) for control group and brain connections in (**C2**) for NFT group and (**D2**) for control group in 2-back condition. The box plots in (**E1**) and (**E2**) show the phase synchronization index in different brain lobes in 1-back and 2-back conditions, respectively. The coloured bars of the connectivity matrices and brain connections represent the strength of synchronization between the EEG channels. It ranges from 0 (no synchronization) to 1 (full synchronization). The diamond shape in the box plots represents the outliers in the data.

## Data Availability

All the data included in this study are available upon request by contacting the corresponding author.

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
