# Peer review of "Exploring the Effects of EEG-Based Alpha Neurofeedback on Working Memory Capacity in Healthy Participants"

_bioengineering, 2023, doi:10.3390/bioengineering10020200_

Round 1

Reviewer 1 Report

In this interesting work Doctor Nawaz and coauthors try to investigate the neural mechanisms underlying working memory before and after alpha band neurofeedback training. I appreciated reading the manuscript, but I think it would take advantage of some refinements and a careful revision. I would like to rise some comments that I hope could be constructive.

Main comments:

-          I think that the rationale and the hypothesis of the study could be better clarified in the introduction and in the discussion, highlighting more which are the main innovative aspects of the study.

-          If I understand correctly, the post-stimulus power in the time-frequency decomposition has been averaged over trials and then normalized to mean baseline values. It would be interesting to verify if the same results can be obtained also performing the normalization procedure at the single trial level, that, as far as I can comprehend, should be more robust.

-          The authors aim to investigate the effects of neurofeedback also in frequency bands others than the one used during the training. However, in the offline analysis of EEG data recorded during the neurofeedback training sessions, only results for the power in the alpha band were included. It would be interesting to include in this section also a description of the power in the theta and beta frequency bands during the session of neurofeedback training (NFT).

-          The phase locking value (PLV) has been used in the study as measure of phase synchronization for functional connectivity analysis. However, several other indexes have been proposed, some also dealing with the problem of volume conduction. For this reason, I think that this choice should be better argued.

Further remarks:

-          In the manuscript is reported that participants were randomly assigned to either the NFT group or the control group. It is not clear if age and gender were however considered in this division to avoid creating bias between the two group. Mean age and gender should however be reported also separately for the NFT and the control group.

-          At line 71 the acronym NFB is used without being announced before.

-          At line 139 is reported that the Welch’s Method was used to compute the absolute value of alpha band power using a 5-sec widow. The type of window used as well as the amount of overlapping between segments should be reported. Moreover, it is not clear how values from single channels have been aggregated in order to obtain an unique value (i.e. simple mean ? ).

-          The precise frequency range considered in the study for both the beta and theta band should be somewhere specified.

-          At line 307 authors affirm that also EEG data collected during the NFT sessions have been analyzed but it is not specified which kind of pre-processing have been applied to these data neither how alpha power has been calculated and which is the baseline considered in this analysis.

-          In the caption of figure 3 should be added a description of outliers as in other figures: “The diamond shape in the box plots represents the outliers in the data”.

-          Results are reported for the frontal, fronto-central, parietal and occipital lobe but I didn’t find any specification regarding the particular EEG channels used for each of the cited analysis. I think this information should be added.

Reviewer 2 Report

You may think of some benchmarking using standard data sets available if any.

Reviewer 3 Report

The manuscript on Neurofeedback traning reveals the functional brain connections in theta band suggested working memory within each hemisphere of the brain is stronger in comparison to the connectivity between the two hemispheres. As suggested by the authors, second control group comparision would extend the result analysis to the realm of clinical usability. It is recommended that patients numbers and control (health individuals) should be of higher numbers for better result, towards clinical usability. 

Reviewer 4 Report

The paper arranged by the authors were fine and good.

need more explanation about the electrode placement.

In figure.1 b diagram was not in clear.

need more explanation for protocol designed.

literature survey need to added in the paper.

Round 2

Reviewer 1 Report

I thank the authors for having taken into consideration all my comments and carefully revised the manuscript. I have not further comments.